# A Broad-Specificity Chitinase from *Penicillium oxalicum* k10 Exhibits Antifungal Activity and Biodegradation Properties of Chitin

**DOI:** 10.3390/md19070356

**Published:** 2021-06-23

**Authors:** Xing-Huan Xie, Xin Fu, Xing-Yu Yan, Wen-Fang Peng, Li-Xin Kang

**Affiliations:** State Key Laboratory of Biocatalysis and Enzyme Engineering, Hubei University, Wuhan 430072, China; aprsunshine@stu.hubu.edu.cn (X.-H.X.); fuxindi2019@163.com (X.F.); yanxingyu20202020@163.com (X.-Y.Y.); wenfang@hubu.edu.cn (W.-F.P.)

**Keywords:** *P. oxalicum*, chitinase, production, chitinous biodegradation, fungicide

## Abstract

*Penicillium oxalicum* k10 isolated from soil revealed the hydrolyzing ability of shrimp chitin and antifungal activity against *Sclerotinia sclerotiorum.* The k10 chitinase was produced from a powder chitin-containing medium and purified by ammonium sulfate precipitation and column chromatography. The purified chitinase showed maximal activity toward colloidal chitin at pH 5 and 40 °C. The enzymatic activity was enhanced by potassium and zinc, and it was inhibited by silver, iron, and copper. The chitinase could convert colloidal chitin to N-acetylglucosamine (GlcNAc), (GlcNAc)2, and (GlcNAc)3, showing that this enzyme had endocleavage and exocleavage activities. In addition, the chitinase prevented the mycelial growth of the phytopathogenic fungi *S. sclerotiorum* and *Mucor circinelloides.* These results indicate that k10 is a potential candidate for producing chitinase that could be useful for generating chitooligosaccharides from chitinous waste and functions as a fungicide.

## 1. Introduction

Chitin, a linear polysaccharide of N-acetylglucosamine (GlcNAc), is the major component of crustacean shell-waste material, insect exoskeletons, and fungal cell walls [1]. It is estimated that 10^14^ tons of chitin are produced every year in the shrimp and crab industry. Due to its highly ordered crystalline structure, chitin is insoluble in common solvents and has a low biodegradation rate by environmental microorganisms; thus, it has a negative impact on the environment, which limits its potential applications in the fields of food, medicine, and agriculture [2,3].

Chitin can be degraded into chitooligosaccharides (COSs), which are water-soluble, biocompatible, and non-toxic and exhibit a broad range of bioactivities, such as food preservation, dietary supplement, wound healing, anti-infection, antimicrobial, antioxidant, and plant growth-promoting properties [4,5,6], making them excellent alternatives. Therefore, the degradation of chitin biomass has become a hot research topic. Chitin can be converted into COSs through either chemical or enzymatic approaches. The enzymatic method, especially using chitinases, has the advantages of being environment friendly, needing mild reaction conditions, and resulting in well-defined COSs with high product safety [7].

Chitinases (EC 3.2.1.14) are hydrolytic enzymes that cleave the β-(1,4)-glycosidic bonds in chitin. These enzymes are widely distributed in fungi [8], bacteria [9], plants [10], and insects [11], and they play important roles to degradate chitinous waste for the production of oligosaccharides and GlcNAc [12,13]. They are also used in agriculture, food, and medicine for the biocontrol of pathogenic fungi [14]. However, it is expensive to use chitinases for the biodegradation of chitin and biocontrol due to its high production cost and low conversion rate [15]. As a result, there is an essential need for better chitinases for the enzymatic hydrolysis of chitin and biocontrol of fungi.

Recently, marine chitinases have attracted attention as one of the potential enzymes. The marine environment of chitinase-producing strains survival includes marine sediments and marine waste deposited soil along the coastal regions. In this study, we isolated the fungus *Penicillium oxalicum* from marine waste deposited soil, which can grow in medium containing shrimp chitin as a sole carbon source and has chitin-degrading activity. In addition, *Sclerotinia sclerotiorum* was used as a model for antagonistic evaluation, which was a devastating fungal plant pathogen with a broad host range (above 408 described species) [16]. Chitinase was optimally produced, and the biological characteristics were examined. Moreover, the applications of bioconversion of chitin and antifungal activity were carried out. Overall, our work aims at studying chitinolytic enzyme characteristics in order to obtain enzyme for industrial applications.

## 2. Results and Discussion

### 2.1. Isolation and Identification of Chitinase-Producing Strains

Chitinase can decompose chitin into simple molecules for cell applications. Furthermore, chitin is one of the major components of the fungal cell, and antifungal activity has been attributed to secreted chitinases lysing the fungal cell wall [17]. In our study, based on the above-mentioned properties, the chitinase-producing strains were isolated. Grown on powder chitin as the sole carbon source, the isolation of potent chitin-degrading strains was performed. Twenty fungus strains have been isolated from the shrimp waste disposal site, five of which showed high chitinase activity. The k10 strain produced the highest chitinase activity (4.2 U/mL). The k10 strain was used to further study the antagonistic property against *S*. *sclerotiorum,* which showed good inhibitory activity (Figure 1). For molecular identification, a 571 bp internal transcribed spacer (ITS) of k10 was sequenced and BLASTed. The results showed that k10 ITS exhibited 100% sequence identity to several *P. oxalicum* strains (GenBank Accession Nos. MT588795.1, MT529129.1, and MT446169.1

### 2.2. Optimization of Culture Conditions for Chitinase Production

To investigate the effects of culture conditions on chitinase production by *P. oxalicum* k10, four chitinous materials and four nitrogen sources were used for fermentation. As shown in Figure 2A, colloidal chitin and powder chitin were the most suitable substrates for chitinase production with high activities. Thus, considering the production cost, powder chitin was chosen as an excellent inducer for further investigation. Figure 2B showed that tryptone was the most suitable nitrogen source for chitinase production. Additionally, basal medium containing 3% powder chitin and 4% tryptone were the most suitable for chitinase production at 28 °C and pH 5 for 3 days (Figure 2C–F). Furthermore, chitinase activity was influenced by carbon supplement in the production medium, which was observed with corn starch, and the activity peaked at 37 U/mL (Figure 2G).

Colloidal chitin was widely used as an inducer for the production of chitinase in various studies. For example, the expression of chitinases from *Serratia marcescens* PRNK-1 [18], *Pedobacter* sp. PR-M6 [19], and *Bacillus licheniformis* B307 [20] were incubated in 0.5% (w/v) colloidal chitin. Chitinases from *Acremonium* sp. YS2-2 [8], *Trichoderma asperellum* PQ34 [21], *Myxococcus fulvus* UM01 [22], and *Achromobacter xylosoxidans* [23] were induced in response to 1% colloidal chitin. The maximum chitinase yield of *Trichoderma viride* [24], *Streptomyces pratensis* KLSL55 [25], and *Hydrogenophilus hirschii* B-DZ44 [26] were obtained at 1.4%, 1.5%, and 2% of colloidal chitin, respectively. Colloidal chitin is a derivative of chitin obtained by acid–base chemical reagent (hydrogen chloride and sodium hydroxide) treatment; as a result, the production cost is high, and it leads to environmental pollution. To solve this problem, powder chitin and shrimp shell have been used to produce chitinolytic enzyme. An appropriate ratio of chitin and colloidal chitin resulted in an enhancement in chitinase production levels, and the optimum concentrations were 7.49 g/L chitin and 4.91 g/L colloidal chitin [27]. *Bacillus altitudinis* produced thermostable chitinase during a 4-day incubation on shrimp shell medium [28]. In our study, powder chitin was chosen as the inducer for chitinase production, which has the advantage of low cost for industrial application.

### 2.3. Chitinase Purification

The chitinase was effectively precipitated from the culture supernatant at a 70% concentration of ammonium sulfate. After the dialysis and Sephadex-75 size-exclusion chromatography, eight fractions containing enzyme activity were obtained. Approximately 190 mg k10 chitinase was obtained and the enzyme specific activity increased to 162.1 U/mg after purification using the Sephadex column (Table 1).

### 2.4. SDS-PAGE and Zymography

In culture supernatants from chitin-grown cells, one major protein band was detected (Figure 3, lane 2), with a molecular weight of approximately 45 kDa (Figure 3, lane 3); while for culture supernatants obtained from no chitin-grown cells, very low signal intensity was shown (Figure 3, lane 1). The chitinase activity of the protein was confirmed with zymography (Figure 3, lane 4).

### 2.5. Effects of Temperature and pH on Chitinase and Kinetic Parameters

After purification, the enzymatic characterization was determined. The purified chitinase exhibited maximum activity (100%) at 40 °C (Figure 4A). The activity of the enzyme remained above 90% for 60 min of incubation at 40 °C but decreased to 70% at 50 °C for 60 min and to 35% at 60 °C for 10 min (Figure 4B). The optimal temperature was similar to other chitinases from *T. asperellum* PQ34 [21], *T. viride* [24], *Chitinibacter Tainanensis* CT01 [29], and *Bacillus subtilis* [30].

K10 chitinase was active in pH ranging between 4 and 7 and was found to have an optimal activity at pH of 5.0 (Figure 4C,D). A similar value was determined for the pH optimum of the chitinases from *H. hirschii* KB-DZ44 [26], *B. subtilis* [30], *Streptomyces albolongus* ATCC 27414 [31], and *Chromobacterium violaceum* [32].

The kinetic values of the chitinase were determined with colloidal chitin substrate, yielding K_m_ of 12.56 mg/mL, k_cat_ of 0.22 s^−1^, and V_max_ of 1.05 μM min^−1^ mg^−1^ protein. An average activity during the 1st h was used to approximate initial activity and the enzyme activity was decreasing by 10% during 1 h assay.

### 2.6. Substrate Specificity of k10 Chitinase

To further investigate the features of the chitinase, substrate specificity was measured. The highest activity was defined as 100%. K10 chitinase had the maximum activity toward colloidal chitin (Table 2). Relative activity on powder chitin, ultrafine chitin, shrimp shell powder, and 90% deacetylated chitosan were 42.6%, 58.3%, 15.7%, and 11.4%, respectively, compared to colloidal chitin.

### 2.7. Effects of Metal Ions and Chemical Reagents on Enzyme Activity

The effects of metal ions and chemical reagents on enzyme activity were determined. As shown in Table 3, the relative enzymatic activity was enhanced by zinc (Zn^2^^+^) and potassium (K^+^), and incubation with silver (Ag^+^), iron (Fe^2^^+^), sodium dodecyl sulfate (SDS), ethylenediaminetetraacetic acid (EDTA), and β-mercaptoethanol decreased the chitinolytic activity to 65.9%, 63.9%, 82.7%, 78.2%, and 75.8%, respectively. The addition of copper (Cu^2^^+^) and carbamide caused a decrease of >50% in chitinolytic activity. In other studies, K^+^ also increased the enzymatic activities from *C. Tainanensis* CT01 [29], *Vibrio harveyi* [33], and rChiT-7 [34]. Similarly, other chitinases were also inhibited by the addition of metal ions and chemical reagents. For example, ChiT-7 activity was suppressed by SDS, EDTA, Cu^2^^+^, and Ag^+^ [34]. *S. marcescens* chitinase was inhibited by Ag^+^, SDS, and Cu^2^^+^ [35]. *Stenotrophomonas rhizophila* G22 chitinase was inhibited by Fe^2^^+^, Cu^2^^+^, and SDS [36]. SaChiA4 activity was inhibited by Cu^2^^+^, EDTA, and SDS [31].

### 2.8. Chitooligosaccharides (COSs) Preparation and Analysis of End Products

To study the effects of enzyme dosage on COSs yield, a varying amount of enzyme from 5 to 20 μg/mL was used. As expected, COSs yield increased with increasing enzyme concentration (Figure 5A). Then, 20 μg/mL chitinase was added to 20 mg/mL colloidal chitin and incubated from 6 to 48 h. After 30 h of treatment, the transparency of the reaction system was significantly increased, and the concentration of COSs peaked at 17 mg/mL (Figure 5B or Figure 6A). The end product of hydrolysis was analyzed by thin layer chromatography (TLC), and the most abundant oligosaccharides were (GlcNAc)2 and (GlcNAc)3, followed by GlcNAc (Figure 6B), indicating that k10 chitinase had endocleavage and exocleavage properties. Thus, k10 chitinase may be an ideal biocatalyst for the utilization of chitin biomass to produce COSs.

Chitinases are classified into two major categories. One category is endochitinases, which cleave chitin to generate soluble low-molecular (GlcNAc)n (n ≥ 2) [37]. For example, chitinases from *Thermoascus aurantiacus, Chaetomium thermophilum,* and *S. marcescens* exhibited an endocleavage pattern and hydrolyzed colloidal chitin to yield mainly (GlcNAc)2 [35,38]. rChit46 from *Trichoderma harzianum* effectively hydrolyzed chitin to produce (GlcNAc)2, (GlcNAc)3, and (GlcNAc)4 [39]. Chitinase from *B. altitudinis* hydrolyzed chitin to produce (GlcNAc)2, (GlcNAc)3, (GlcNAc)4, and (GlcNAc)5 [40]. Another chitinase category is exochitinases (beta-glucosaminidases and chitobiosidases), which cleave chitin or (GlcNAc)n to generate GlcNAc or (GlcNAc)2. For example, chitinases from *Aeromonas caviae* CHZ306 [41] and *B. licheniformis* AT6 [42] produced GlcNAc from chitin. Chitinases from *Chitiniphilus shinanonensis* [43] and *Eiseniafetida* [44] efficiently hydrolyzed colloidal chitin to release (GlcNAc)2.

However, at present, a few chitinases have the simultaneous catalytic activities of endochitinase and exochitinase, which are the so-called broad-specificity chitinases [45]. For example, chitinases from *Paenibacillus timonensis* LK-DZ15 [46] and *Bacillus* sp. DAU101 [9] cleaved colloidal chitin into a mixture of (GlcNAc)2, (GlcNAc)3, (GlcNAc)4, (GlcNAc)5, and GlcNAc. Chitinases from *Humicola grisea* [27] and *Saccharothrix yanglingensis* Hhs.015 [47] were capable of hydrolyzing chitin to (GlcNAc)3, (GlcNAc)2, and GlcNAc. Our results were similar to those of broad-specificity chitinases, and they showed endocleavage and exocleaveage activity, degrading colloidal chitin to produce (GlcNAc)3, (GlcNAc)2, and GlcNAc. Based on the catalytic mechanism, broad-specificity chitinases are more effective than single-activity chitinases, with obvious advantages of low cost and high catalytic efficiency on the degradation of chitin into COSs, which are good candidates for the green conversion of chitinous waste and have great valuable in the aspects of research and application.

### 2.9. Antifungal Activity of Chitinase

The antifungal activity of k10 chitinase against phytopathogenic fungi *S. sclerotiorum* and *M. circinelloides* was tested by the hyphal extension assay. As present in Figure 7, the inhibitory activity of k10 chitinase was observed compared to the control, which exhibited clear inhibitory zones of mycelial growth against *S. sclerotiorum* and *M. circinelloides.* These results indicate that the antagonistic property of k10 strain against *S. sclerotiorum* is related to its production of chitinase.

Since chitin is the key component of fungal cell walls, chitinases play an important role in plant defense against fungal pathogens. The antifungal activity of chitinases have been observed in diverse studies (Table 4). Similar to the reported chitinases, the antifungal activity of chitinase from *P. oxalicum* makes it a bioactive material for pathogenic fungi control in plant diseases and harmful food fungi.

## 3. Materials and Methods

### 3.1. Chemicals

Shrimp shell chitin and chitosan were purchased from Sigma-Aldrich (St. Louis, MO, USA). (GlcNAc)n (n = 1–4) were purchased from BZ Oligo Biotech (Qingdao, Shandong Province, China). All other chemicals and solvents were acquired from local suppliers and were of analytical grade and high purity.

### 3.2. Preparation of Colloidal Chitin and Ultrafine Chitin

Colloidal chitin was prepared according to the method by Sandhya et al. [55] with a few modifications. Briefly, shrimp shell chitin and hydrogen chloride (37%, *m*/*v*) were mixed at a 1:5 ratio (*w*/*v*). After incubation at 4 °C overnight, the pH was adjusted to neutral with sodium hydroxide, tap water was added to the viscid solution, and the colloidal chitin was separated by centrifugation at 10,000 *g* and stored at 4 °C. Ultrafine chitin was prepared according to the method by Pareek et al. [56]. Chitin was dissolved in methanol with calcium chloride dihydrate overnight. Dissolved chitin was precipitated using calcium citrate solution; then, it was dialyzed and dried. Ultrafine chitin thus obtained was further treated with formic acid and dried.

### 3.3. Screening, Cultivation, and Identification of the Microbial Strain

The strains were isolated from soil from a shrimp waste disposal site surrounding a marine beach in Beihai Guangxi, China. The samples were inoculated on chitin powder (as a sole carbon source) agar medium containing powder chitin 10 g/L, (NH_4_)_2_SO_4_ 1 g/L, K_2_HPO_4_ 2 g/L, KH_2_PO_4_ 1 g/L, NaCl 0.5 g/L, MgSO_4_ 0.5 g/L, and agar 20 g/L, and incubated at 30 °C for 5 days. The strains that grew vigorously were picked. To evaluate the hydrolysis ability of chitin, strains were inoculated in medium containing chitin powder as a sole carbon source 10 g/L, tryptone 5 g/L, KH_2_PO_4_ 1 g/L, K_2_HPO_4_ 1 g/L, MgSO_4_ 0.5 g/L, FeSO_4_ 0.25 g/L, and NaCl 1 g/L, and then were cultivated for 120 h in a shaking incubator (200 rpm) at 28 °C. The supernatant was collected by centrifugation every 24 h during the 5-day cultivation period, and the chitinase activity was measured. To identify the fungi, a polymerase chain reaction (PCR) was performed to amplify the ITS gene from the genomic DNA. The forward primer was 5′-TCCGTAGGTGAACCTGCGG-3′, and the reverse primer was 5′-TCCTCCGCTTATTGATATGC-3′. The temperature cycle was at 94 °C for 30 s, 56 °C for 30 s, and 72 °C for 45 s for 35 cycles and 8 min at 72 °C for extension. The nucleotide sequence alignments were conducted using the BlastX program of NCBI (http://www.ncbi.nlm.nih.gov/BLAST (accessed on 24 November 2020)).

### 3.4. Antagonistic Activity Evaluation of Strain


For the evaluation of antagonistic activity, the S. sclerotiorum was preliminarily streaked on a potato dextrose agar (PDA) plate and incubated at 28 °C for 3 days. Then, 0.8 cm of the fungus colony on the PDA plate was cut and streaked on the center of a new PDA plate containing 1% colloidal chitin. The spore of the target strain was added to the agar plate, at a distance 1.5 cm away from the rim of the S. sclerotiorum colony. The plate was incubated at 28 °C and was observed for the appearance of a zone of fungal growth inhibition.

### 3.5. Determination of Chitinase Activity and Oligosaccharide Content

Chitinase activity was determined by the release of GlcNAc equivalents from colloidal chitin with a GlcNAc standard. The activity of chitinase on colloidal chitin was measured using the 3,5-dinitrosalicylic acid (DNS) assay [18] comprising 800 μL 1% colloidal chitin in 100 mM sodium acetate buffer (pH 5) and 200 μL enzyme. The mixture was incubated at 37 °C for 1 h. Then, the reaction was terminated by adding 1 mL DNS reagent and heating in boiling water for 10 min. The supernatant was collected by centrifugation. Next, 0.5 mL supernatant was mixed with 0.5 mL DNS and incubated at 100 °C for 10 min. The oligosaccharide content was estimated at 542 nm. One unit of chitinase activity was defined as the amount of enzyme that released one micromole of GlcNAc equivalent per min at 37 °C.

### 3.6. Optimization of Chitinase Production

To maximize chitinase production, the medium and process were optimized for shake flask cultures by one factor at a time. The strain was grown in basal medium (K_2_HPO_4_ 5 g/L, MgSO_4_ 2 g/L, FeSO_4_ 1 g/L, NaCl 5 g/L), the induction of chitinase was checked by incorporating 1% inducer (colloidal chitin, powder chitin, shrimp shell powder, and chitosan), and the enzyme production was determined. Different nitrogen sources such as tryptone, peptone, beef extract, and soybean meal powder were chosen to compare chitinase production. The inducer and nitrogen source, which supported the maximum production of chitinase, were further tested at different concentrations for enhancement of enzyme production. The effects of pH and temperature were investigated by cultivating the test isolate in production medium at various pH (2–9) and temperature (24–34 °C) ranges. Carbon sources such as glucose, sucrose, cornstarch, and yeast extract were supplemented with the production medium to study their influence on chitinase activity.

### 3.7. Enzyme Purification

The cultures were centrifuged at 10,000 *g* for 10 min at 4 °C to obtain the supernatant, which served as the crude enzyme. The crude enzyme was precipitated with a 70% concentration of ammonium sulfate slowly added to the supernatant under constant stirring at 4 °C. The precipitate was collected by centrifuging at 10,000× *g* for 30 min at 4 °C and resuspended in Tris-HCl buffer (10 mM, pH 6.0). It was dialyzed against the same buffer for 24 h at 4 °C. Then, the dialyzed enzyme solution was purified with Sephadex 75 column (20 cm × 1.5 cm) pre-equilibrated with 3 column volumes of Tris-HCl buffer (10 mM, pH 6.0). The flow rate of 0.3 mL/min was used to elute the enzyme, and 0.6 mL/fraction/2 min was collected; in this way, 20 fractions were collected. The eluted fractions were assayed for chitinase activity. Fractions showing high chitinase activity were concentrated using a 30 kDa centricon Plus-20 (MILLIPORE). The purified enzyme was analyzed by gel electrophoresis and zymography, and the biochemical properties were determined.

### 3.8. Sodium Dodecyl Sulfate-Polyacrylamide Gel Electrophoresis and Zymography

Chitinase was resolved by sodium dodecyl sulfate-polyacrylamide gel electrophoresis (SDS-PAGE), and the protein bands were visualized with Coomassie brilliant blue R-250. Zymography was performed using the method by Yuli [57]. Briefly, chitinase protein, no cooking, and 2-mercaptoethanol were separated on a 12% polyacrylamide gel by incorporating 0.1% colloidal chitin. After electrophoresis, SDS removal, and rinsing, the gel was incubated in 50 mM Tris-HCl buffer (pH 5) at 37 °C for 12 h to allow the chitinase to hydrolyze the colloidal chitin substrate, and it was subsequently stained with 0.1% of Congo red solution followed by washing with 1 M NaCl. Then, the lytic zone was visualized.

### 3.9. Effects of Temperature and pH on Chitinase Activity and Stability

The temperature-dependent activity profile of purified chitinase activity was determined in 50 mM acetate buffer (pH 5.0) at temperatures from 30 to 60 °C. The pH activity profile of chitinase was determined at 40 °C and pH 3.0–10.0. For the thermal stability assay, the purified enzyme was pre-incubated at 40 °C, 50 °C, and 60 °C for 10–60 min, followed by the residual activity assay toward colloidal chitin. pH stability was determined by pre-incubating the purified enzyme at various pH using the same buffer system at 4 °C for 24 h, followed by the residual activity assay under standard conditions.

### 3.10. Kinetic Analysis of k10 Chitinase

Kinetic parameters of k10 chitinase were determined by incubating 1.0 μM of enzyme with different concentrations of colloidal chitin (0–20 mg/mL) in 100 mM sodium acetate buffer (pH 5) with respective controls in triplicates at 40 °C for 1 h. The reducing ends generated were measured by chitinase assay as described in Section 3.5. Kinetic parameters were evaluated from three independent sets of data fitting to the Michaelis–Menten equation by nonlinear regression function using GraphPad Prism version 5.0 (GraphPad Software, San Diego, CA, USA).

### 3.11. Substrate Specificity Determination

The substrate specificity of purified k10 chitinase was tested under standard conditions (40 °C, pH 5.0 for 1 h) with 1% (*w*/*v*) of colloidal chitin, shrimp chitin powder, ultrafine chitin, shrimp shell powder, 90% deacetylated chitosan, and cellulose as the substrate.

### 3.12. Effects of Metal Ions and Chemical Reagents on Enzyme Activity

The effects of metal ions and chemical reagents on chitinase were investigated by measuring the enzyme activity in the presence of 10 mM K^+^, Mg^2^^+^, Fe^2^^+^, Cu^2^^+^, Na^+^, Ca^2^^+^, Zn^2^^+^, Ag^+^, carbamide, EDTA, β-mercaptoethanol, and SDS. The activities of untreated enzyme were defined as 100%.

### 3.13. Preparation of COSs

To study the best combination of the reaction time and enzyme dosage, a 2 mL reaction mixture consisting of 5–20 μg/mL enzyme for each experimental and 20 mg/L colloidal chitin in 100 mM sodium acetate buffer (pH 5.0) was incubated at 40 °C. Aliquots were analyzed for hydrolyzed products at different time intervals.

### 3.14. Thin Layer Chromatography Analysis of end Products of Hydrolysis

The end products of hydrolysis were separated on a glass-backed TLC Aluminum Silica Gel Plate (Merck, Darmstadt, Germany) in a solvent system containing iso-propanol/ethanol/water/ammonia at the ratio of 5:5:4:0.3 (*v*/*v*/*v*/*v*) as the mobile phase. (GlcNAC)n (n = 1–4) was used as the standard. The plate was dried and sprayed with 0.5% ninhydrin reagent (0.5 g ninhydrin dissolved in 100 mL ethanol), followed by incubation at 115 °C to develop the signal.

### 3.15. Assay for Antifungal Activity of Chitinase

The antifungal activities of k10 chitinase were tested by employing a hyphal extension inhibition assay as previously described [58]. Then, 0.8 cm of the *S. sclerotiorum* and *M. circinelloides* fungal colonies were transferred to the PDA plate and incubated until fungal growth was observed. Sterile oxford cups were placed on the plate at a distance of 1.5 cm away from the rim of the mycelial colony, after which purified chitinase (filtered with pore size of 0.22 um) and control (thermo-denatured chitinase) were added. The plates were incubated at 28 °C until mycelial growth had enveloped the wells containing formed zones of inhibition around the cups containing chitinase with antifungal activity.

### 3.16. Statistical Analysis

All data were shown as the mean of at least three independent replicates with their standard deviations. GraphPad Prism 5.0 was used for statistical analysis, and the treatments with a *p* value of less than 0.05 were considered to be statistically significant.

## 4. Conclusions

A *P. oxalicum* k10 strain capable of degrading chitin and showing chitinase activity was isolated. The production of chitinase by k10 degrading powder chitin was demonstrated. The K^+^ and Zn^2+^ were promoters of chitinase activity, and Cu^2+^, Fe^2+^, Ag^+^, carbamide, EDTA, β-mercaptoethanol, and SDS were inhibitors. TLC results showed that k10 chitinase had endocleavge and exocleavage properties. The major enzymolysis end products were (GlcNAc)2, (GlcNAc)3, and a small amount of GlcNAc. Furthermore, the chitinase exhibited clear inhibitory of mycelial growth against *S. sclerotiorum* and *M. circinelloides.* These results reveal that *P. oxalicum* chitinase is a good candidate as a bioconversion agent of chitinous waste and can be exploited as a fungicide.

## Figures and Tables

**Figure 1 marinedrugs-19-00356-f001:**
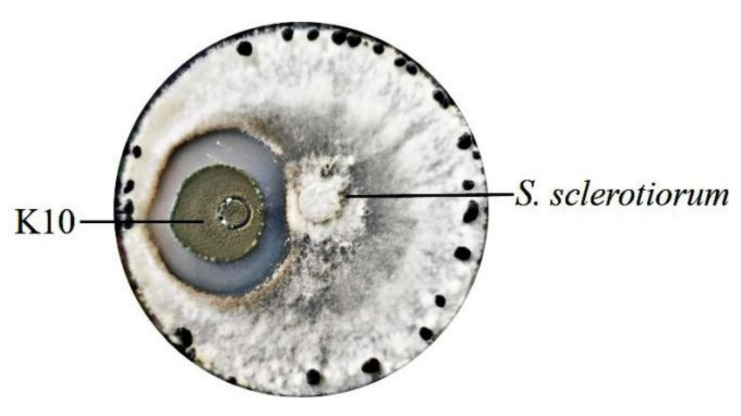
Antagonistic property of k10 strain against the hyphal growth of *S. sclerotiorum* by a culture assay. The spore of the k10 strain was inoculated at a distance 1.5 cm away from the rim of the *S. sclerotiorum* colony and incubated at 28 °C.

**Figure 2 marinedrugs-19-00356-f002:**
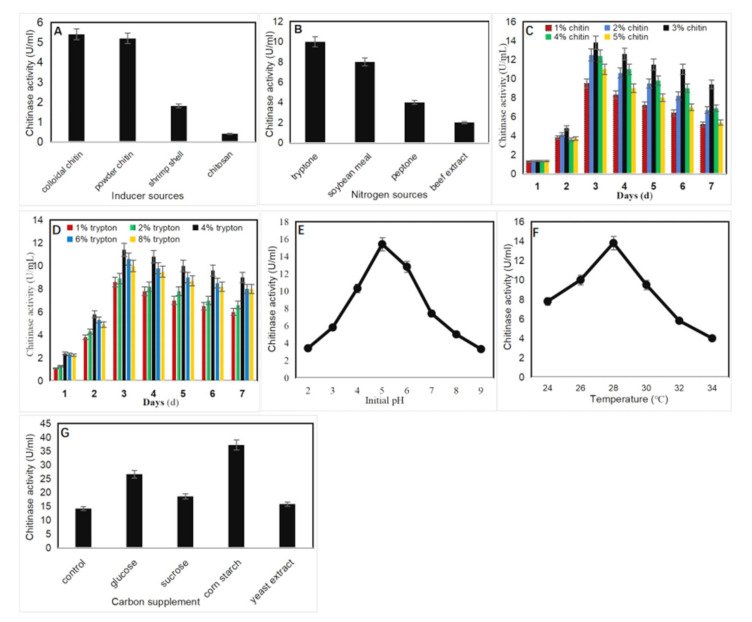
Effects of culture conditions on chitinase production by *P. oxalicum* k10: (**A**) Different inducer sources on chitinase production; (**B**) Different nitrogen sources on chitinase production; (**C**) Different concentrations (1–5%) of powder chitin on chitinase production at pH 6; (**D**) Different concentrations (1–8%) of tryptone on chitinase production at pH 6; (**E**) Effects of pH (2–9) on chitinase production at 28 °C; (**F**) Effects of temperature (24–34 °C) on chitinase production at pH 5; (**G**) Carbon sources supplement on chitinase production at 28 °C and pH 5.

**Figure 3 marinedrugs-19-00356-f003:**
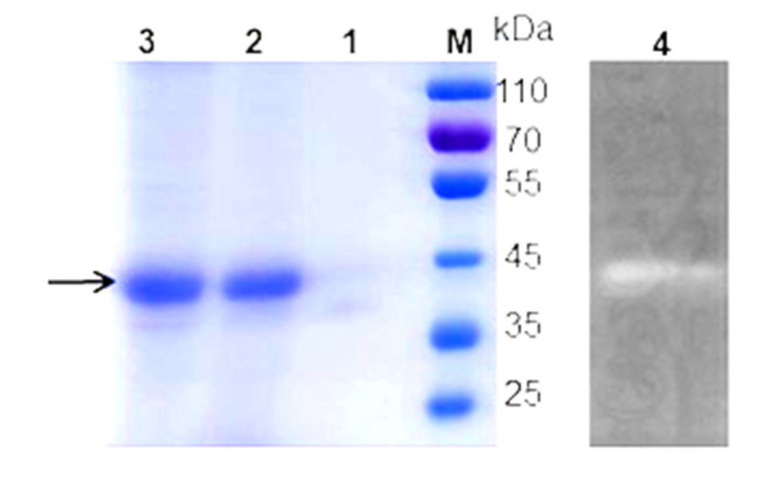
SDS-PAGE and zymography analyses of chitinase. Lanes: M, standard protein molecular weight markers; 1, culture supernatant of the control strain (no chitin inducer); 2, culture supernatant of the crude chitinase (2.6 μg); 3, purified chitinase (3.0 μg); 4, zymogram of the purified chitinase (2.0 μg).

**Figure 4 marinedrugs-19-00356-f004:**
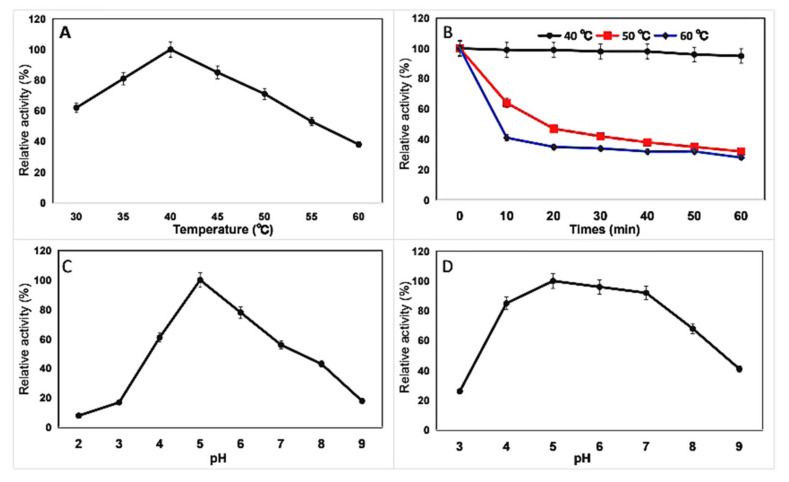
Enzymatic features of chitinase: (**A**) Temperature effects of chitinase toward colloidal chitin; (**B**) Thermostability of chitinase; (**C**) The pH effects of chitinase; (**D**) pH stability of chitinase. The data were presented as mean ± standard deviation of three biological replicates.

**Figure 5 marinedrugs-19-00356-f005:**
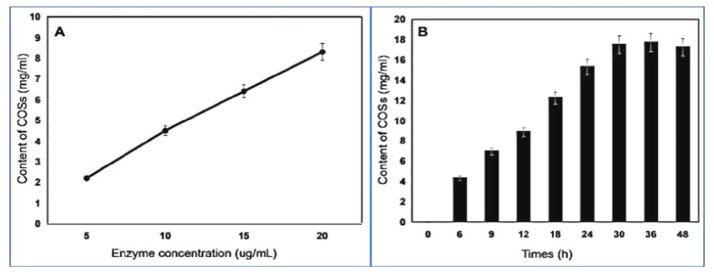
(**A**) Yield of COSs with 5 to 20 μg/mL chitinase with 12 h treatment; (**B**) Yield of COSs from 20 mg/mL colloidal chitin degradation with 20 μg/mL chitinase added for 6 to 48 h.

**Figure 6 marinedrugs-19-00356-f006:**
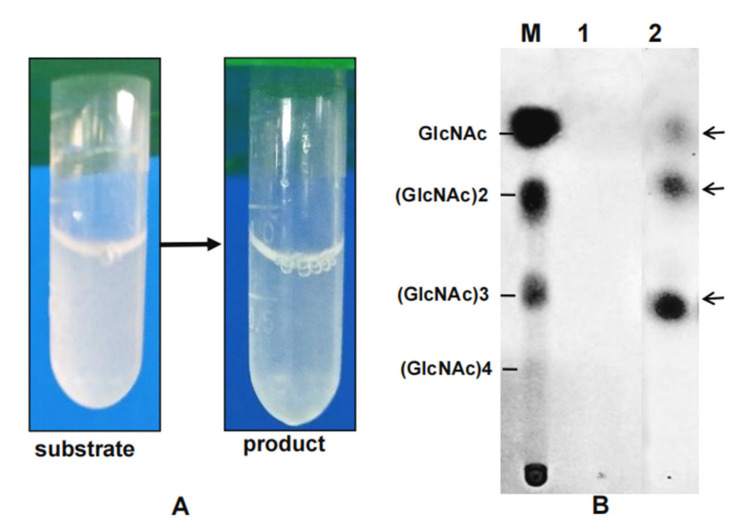
(**A**) Transparency of the reaction system between substrate and product (30 h treatment); (**B**) Analysis of the hydrolytic products of colloidal chitin by TLC. Lane M, the standard of (GlcNAc)n (n = 1–4); 1, control of chitin; 2, hydrolysate of chitin with 36 h treatment.

**Figure 7 marinedrugs-19-00356-f007:**
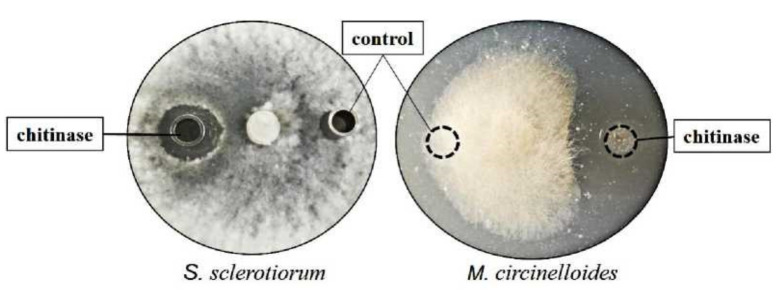
Antifungal activity of purified enzyme produced from k10 against *S. sclerotiorum* and *M. circinelloides* after incubation on PDA medium. Control, thermo-denatured chitinase. Purified chitinase (2 μg) and control were added at a distance of 1.5 cm away from the rim of the mycelial colony, respectively, and incubated at 28 °C.

**Table 1 marinedrugs-19-00356-t001:** Estimation of *P. oxalicum* chitinase activity.

Steps	Total Protein (mg)	Total Activity (U)	Specific Activity (U/mg)	Purification Fold	Yield (%)
Culture Supernatant	265	37,000	139.62	1	100
(NH_4_)_2_SO_4_ Precipitate	220	33,750	153.41	1.1	83
Sephadex Column Purified	190	30,800	162.1	1.16	72

**Table 2 marinedrugs-19-00356-t002:** Substrate specificity of purified k10 chitinase.

Substrates	Relative Activity (%)
Colloidal Chitin	100.0 ± 1.2
Powder Chitin	42.6 ± 0.8
Ultrafine Chitin	58.3 ± 1.4
Shrimp Shell	15.7 ± 1.0
90% Deacetylated Chitosan	11.4 ± 0.6
Cellulose	0

**Table 3 marinedrugs-19-00356-t003:** Effects of different metal ions and chemical reagents on the purified chitinase ^a^.

Additives	Relative Activity (%)
Control	100.0 ± 1.8
Zinc (Zn^2+^)	120.5 ± 3.1
Potassium (K^+^)	185.6 ± 4.2
Sodium (Na^+^)	101.9 ± 1.8
Calcium (Ca^2+^)	95.3 ± 2.7
Magnesium (Mg^2+^)	88.8 ± 3.5
Silver (Ag^+^)	65.9 ± 2.6
Iron (Fe^2+^)	63.9 ± 3.4
Copper (Cu^2+^)	40.3 ± 2.3
SDS	82.7 ± 3.9
EDTA	78.2 ± 3.6
Carbamide	49.5 ± 1.8
β-Mercaptoethanol	75.8 ± 3.0

^a^ Values represent means ± SD (n = 3) relative to the untreated control samples.

**Table 4 marinedrugs-19-00356-t004:** Antifungal activity of chitinases toward various phytopathogenic fungus.

Chitinase Sources	Phytopathogenic Fungus	References
*P. oxalicum* k10	*S. sclerotiorum* and *M. circinelloides*	This study
*A. griseoaurantiacus*	*F. solani*	[48]
*T. harzianum*	*B. cinerea*	[39]
*T. viride*	*F. oxysporum* f. sp. lycopersici race 3	[24]
*W. anomalus* EG2	*F. oxysporum* KACC 40032 and *R. solani* KACC 40111	[49]
*B. altitudinis*	*V. Candidiasis* and *S. Fungi*	[28]
*B. Pumilus* MCB-7	*A. flavus, A. niger, A. fumigatus,* *C. hydrophila,* and *F. oxysporum*	[50]
*P. elgii* HOA73	*Cladosporium, B. cinerea*	[51]
*Salinivibrio* sp. BAO-1801	*F. oxysporum* and *R. solani*	[52]
*S. galilaeus* CFFSUR-B12	*M. fijiensis* Morelet	[53]
*P. ussuriensis* Maxim	*F. oxysporum, F. solani, R. solani,* and *T. viride*	[54]

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
