# Peer review of "A Broad-Specificity Chitinase from *Penicillium oxalicum* k10 Exhibits Antifungal Activity and Biodegradation Properties of Chitin"

_marinedrugs, 2021, doi:10.3390/md19070356_

Round 1

Reviewer 1 Report

The manuscript entitled: “A Broad-specificity Chitinase from Penicillium oxalicum k10 Exhibits Antifungal Activity and Biodegradation Properties of Chitin”, reference: 1249389

General comments:

There are clear improvements of the clarity and quality of the manuscript, nevertheless there are still some important issues:

What type of statistical analysis were performed by the authors?

I could not clearly find the results mentioned in section 3.10 (line 352 to 358), can the authors please help me? In addition, I would like to analyse the Michaelis-Men-357 ten equation, but I could not find it.  

Please carefully revise all acronyms and their adequate definition and consistency.

Minor comments:

Line 11, “activity of” please consider replacing by “activity against”.

Line 285, please adequately define acronym ITS.

Reviewer 2 Report

The paper "A Broad-specificity Chitinase from Penicillium oxalicum k10 Exhibits Antifungal Activity and Biodegradation Properties of Chitin" provides a study of a newly discovered wide-specificity fungal chitinase. 

I can see that the paper underwent some corrections already. It is generally ok and with some slight corrections might be published (see details in the attachement). I have found one error for which however it is too late to correct. The authors should describe it clearly in the text. Here I will write it once again as it is very important not to propagate this inaccurate notion. The authors can be to some extent absolve from the blame as this incorrect notion is repeated also in many biochemistry books on enzymology:

In the temperature dependence of enzyme activity the maximum of the curve IS NOT AN OPTIMAL TEMPERATURE of enzyme. Quite the opposite it is the T at which temperature denaturation starts to overcome the Arrhenius speed-up of the chemical reaction by increased T. Therefore, all activity assays and characterizations should be conducted at lower T (30oC would be ok).

The way to detect optimal activity is what authors did - thermal stability or long term reaction assays sampled at different time points (reactor tests). 

The authors should get this information into their mind and never ever write again such enzymological nonsense. 

Round 2

Reviewer 1 Report

The manuscript entitled: “A Broad-specificity Chitinase from Penicillium oxalicum k10 Exhibits Antifungal Activity and Biodegradation Properties of Chitin”, reference: marinedrugs-1249389

I would like to congratulate the authors for their efforts. In my opinion the manuscript impact and quality was clearly enhanced.

Author Response

This manuscript is a resubmission of an earlier submission. The following is a list of the peer review reports and author responses from that submission.

Round 1

Reviewer 1 Report

The manuscript entitled: “Production, Purification, and Characterization of A Fungal Chitinase Showing the Antifungal and Biodegradation Properties of Penicillium oxalicum k10”, reference: 1174655.

General and major comments:

The manuscript importance is unquestionable, and is well conceptualized and written. Nevertheless, in my opinion, the manuscript deserves an enhancement of its impact and clarity by a more detailed description of the information in several sections. Also, the authors clearly wrote the text considering the Material and Methods previous to the Results section. The authors should carefully revise this, since the readers now read it in the “opposite direction”: results first and then the materials and methods.

Furthermore, in my opinion the manuscript would considerably improve its impact if the authors determined the Michaelis-Menten kinetics of the chitinase, discribing all its parameters.

Finally the authors state in line: ”we isolated the fungus Penicillium oxalicum from soil”, are the authors absolutely certain of this? Did you perform the complete sequencing? Fortunately, this is not stated in the manuscript text, only in the Abstract section. Please comment.

Point by point suggestions:

Units must be separated from its numerical value. Please revise carefully throughout the entire manuscript. In addition, in line 257, the gravitational force in g, the g must be italicized, in order to be distinguishable from the unit grams

Line 50, please consider adding the term “source” in “carbon source”. The same suggestion for line 262.

Line 62, the first mention to Sclerotinia sclerotiorum should include its full name, and afterwards should be consistently synthesized to S. sclerotiorum. Furthermore, in my understanding the use of this fungi as model for the antagonistic evaluation should be justified and contextualized in the Introduction section.

Line 86, please revise example typo.

Line 90, the first reference to all species must be as their full name. Please carefully revise throughout the manuscript.

Line 92, the statement: “Colloidal chitin is a derivative of chitin obtained by acid-base chemical reagent treatment; as a result, the production cost is high and it leads to environmental pollution” in my opinion is oversimplified. Please provide further details on the definition of colloidal chitin, as well as how it leads to environmental pollution. Please support all the information with adequate references.

Line 98 to 100, is the shell shrimp less advantageous for industrial application then powder chitin? Please provide additional information to support this choice.

Figure 2 is sometimes hard to follow can the authors please add colour?

Table 1 is cropped. Please revise.

Line 158, all acronyms must be adequately defined previously to their use. Please revise throughout the manuscript. In my opinion it would be a plus if the authors also defined the ions names prior to their chemical formula.

Table 3 is cropped, furthermore I would like to understand the value of the control. Can you please explain?

Figure 6 B, the marker molecular weight is missing.

Table 4 is cropped.

Line 258, please briefly describe the method reported by Pareek et al.

Reviewer 2 Report

Here in this manuscript the authors investigated the production, purification and characterization of a fungal chitinase. In general the manuscript is of interest, but contains a lot of drawbacks, which does not justify publication at the moment.

critics:

i) Please specify in more details the results about "Twenty fungus strains were isolated from the shrimp waste disposal site, of which five showed high chitinase activity. The k10 strain produced the highest chitinase activity (4.2 U/mL). The k10 strain was used to further study the antagonistic properties
against S. sclerotiorum, which showed good inhibitory activity...j"

Please include a table concerning all investigated strains. 

ii) What is the treshold of chitinase activity? Not clear. Why is 4.2U/ml high? and not low? Biological relevance is missing. Must be improved for a better understanding.

iii) In figure 2 a chitinase acitive of >5 U/ml) was detected. This is not in accordance with the statement given in line 63! Unclear. 

iv) figure 2: Which ph-value was used in fig. 2D compared to fig 3E or fig 3F? Which ph-value was used in case of 28°C temperature ? Are the ph-values in line with the temperature used in fig. 3F. Unclear. Not described in the figure legend. In general the quality of the labeling/format/style of figure 3 is not ready for publication. The symbols of 3D, 3C are not readable. 

v) Format and style of table 1 must be revised. Some symbols are not legible.

vi) figure 3: Where do you know, that chitinase does have a molecular weight of 45kD? How much did you apply on the SDS-Page in µg/ml? Not given in the figure legend. I'm really wondering why the signal intensity in lane 3 is very low? Must be explained.

vii) In all cases the figure legends must be improved, because it is not clear what is shown, like mean value, median , standard deviation. Number of independent experiments. All figures must be self-explanatory. This is not the case here. 

viii) figure 4: How many values per each condition were measured? Not given. What is shown here? Mean value, standard deviation? Unclear. Why did you combine values by a line? Why no trendlines or regression lines were used in combination with appropriate formula? Not clear. 

ix) table 2: There is a number given "154" ??? Why? Not clear. Furthermore in table 3 the style and  format must be improved. How is it possible to get an activity >100% ? The control shows an activity of 1100 %. How is this possible. A mathematical explanation must be given by the authors.

x) "COS": Definition? What is COS? Not given in section 2.5

xi) figure 5: Which treatment time was used in figure 5A? This must be inserted compared to figure 5B. Not clear. 

xii)  figure 6: figure legend must be improved. What is shown in lane "M"? Why not signal was detected in lane 1? 

xiii) figure 7: What is meant here by "control"? 

xiv) In materials and methods: I miss detailed information about the number of values, the number of independent experiments, the number of samples per each parameter tested. No information about the calculation. Mean values, medians, trend lines, regression lines.........

Reviewer 3 Report

The study of Xie at al. is related on the production and characterization of a fungal chitinase from Penicillium oxalicum. Although the study seems to be interesting, after careful reading I found a few major concerns:

  • What is novelty of this study? What is the scientific contribution in relation to thousands of other similar papers published elsewhere in the literature?
  • Except of the biochemical characterization of the isolated and purified enzyme, what is its practical utility? Today, this filed is rather mature. It must be demonstrated at much higher extend beyond the simple in vitro antagonistic assay with two phytopathogenic fungal strains.

Except of these major concerns, this manuscript needs serious English proofreading.  The manuscript title is not appropriate and have to be changed. Also, their no data regarding:

1) How chitinolytic strains have been isolated,

2) How ITS identification was performed – which primers, PCR and sequencing PCR conditions have been used? What is a rationale to claim that K10 strain is Penicillium oxalicum, regarding its molecular-genetic identification based on the ITS sequence.

3) ITS sequence has to be deposited in the NCBI GenBank data base.

Regarding the chitinolytic strains isolation authors declared: The strains were isolated from soil from a shrimp waste disposal site surrounding marine beach in Beihai Guangxi, China, but did not describe any relevant data regarding the isolation procedure, the cultivation media used for the targeted fungi isolation, so overall procedure is missing; this is a mandatory step,  and must be described in detail.

Also, in the Section 2.1. Isolation and identification of chitinase-producing strains, there is no data on the cultivated collection, neither their chitinolytic potency have been represented.

Likewise, the section 3.3 is named Screening, identification, and cultivation of the microbial strain, while there is no data regarding strain(s) identification, and the cultivation step is poorly described. Also, the antagonistic activity (not antifungal activity as was mentioned by authors) is described within this chapter opposite to the section name. Overall text should be divided into two sections: 1) Screening, cultivation and identification and 2) Antagonistic activity evaluation, containing all relevant mythological steps.

What is pH value of the basal liquid media containing chitin powder used for the induction of the chitinase production?

There is a grammatical error within the sentence in the line 60-61: Grown with powder chitin as the sole carbon source, the isolation of potent chitin-degrading strains. Please, reformulate it.

Line 62: Twenty fungus strains were isolated, should be twenty fungal strains have been isolated...

Line 63: of which five showed high chitinase activity, should be five of which showed...

Lines 95-96: put the organism name for this data.

Line 136: ..was found to have an optimum pH of 5.0 – should be an optimal activity at pH of 5.0

Line 141: The highest activity was defined as 100%. However, the highest activity has been recorded in the presence of shrimp shell and was 115.7%, as shown in Table 2. So this seems to be contradictory. On the other size, data in the text (line 144) is 15.7%, so what is a right data?

Authors should provide any explanation regarding the similar activity on powder and colloidal chitin, because this is not often phenomenon.

The images in Figure 2 should be separated into three rows. This is very difficult to follow and read data presented on this way.

The highest temperature in Image F (Fig. 2) is 34C, not 35C as was stated in the text bellow Fig. 2.

There is an issue regarding data presented within all tables. Some data are not visible – this must be resolved.

The original polyacrylamide gel of the SDS-PAGE chitinase analysis has to be shown. The zymogram image is not visible, and much better image has to be provided.

The control chitinase activity in Table 3 was 1100%?